# Dyslipidaemia Is Associated with Severe Disease Activity and Poor Prognosis in Ulcerative Colitis: A Retrospective Cohort Study in China

**DOI:** 10.3390/nu14153040

**Published:** 2022-07-24

**Authors:** Zhaoshi Liu, Hao Tang, Haozheng Liang, Xiaoyin Bai, Huimin Zhang, Hong Yang, Hongying Wang, Li Wang, Jiaming Qian

**Affiliations:** 1Department of Gastroenterology, Peking Union Medical College Hospital, Chinese Academy of Medical Sciences and Peking Union Medical College, Beijing 100730, China; pumc_liuzhaoshi@student.pumc.edu.cn (Z.L.); tanghao@pumch.cn (H.T.); pumc_lianghz@student.pumc.edu.cn (H.L.); baixiaoyin@pumch.cn (X.B.); huiminzhang@zzu.edu.cn (H.Z.); yangh@pumch.cn (H.Y.); 2State Key Laboratory of Molecular Oncology, National Cancer Center/National Clinical Research Center for Cancer/Cancer Hospital, Chinese Academy of Medical Sciences and Peking Union Medical College, Beijing 100021, China; hongyingwang@cicams.ac.cn; 3Department of Epidemiology and Biostatistics, Institute of Basic Medical Sciences Chinese Academy of Medical Sciences, School of Basic Medicine Peking Union Medical College, Beijing 100005, China; liwang@ibms.pumc.edu.cn

**Keywords:** ulcerative colitis, abnormal serum lipid, lipid profiles, inflammation, disease severity, surgery, survival analysis, nutrition

## Abstract

Background: Clinical data on the correlation of dyslipidaemia with the long-term outcomes of ulcerative colitis (UC) are limited. This study aimed to evaluate the impact of lipid levels on disease activity and prognosis in UC. Methods: The retrospective data of UC patients who had detailed lipid profiles were collected from January 2003 to September 2020. All patients were followed-up to 30 September 2021. The long-term outcomes were UC-related surgery and tumorigenesis. Results: In total, 497 patients were included in the analysis. Compared to patients with normal lipid levels, those with dyslipidaemia commonly presented with more serious disease activity. Low high-density lipoprotein cholesterol (*p* < 0.05) levels were associated with higher risks of severe disease activity in UC. Regarding the long-term outcomes, patients with persistent dyslipidaemia were at higher risks of UC-related surgery (HR: 3.27, 95% CI: 1.86–5.75, *p* < 0.001) and tumorigenesis (HR: 7.92, 95% CI: 3.97–15.78, *p* < 0.001) and had shorter surgery- and tumour-free survival (*p* < 0.001) than patients with transient dyslipidaemia and normal lipid levels. Low levels of high-density lipoprotein cholesterol (*p* < 0.001) and apolipoprotein A1 (*p* < 0.05) were associated with higher risks of surgery and tumorigenesis. Conclusion: Persistent dyslipidaemia was associated with a higher risk of serious disease activity and worse long-term outcomes among patients with UC. Lipid patterns should be assessed to improve the management of high-risk patients with UC in the early phase.

## 1. Introduction

Ulcerative colitis (UC) is a subtype of inflammatory bowel disease (IBD) characterized by inflammation of the gastrointestinal tract with a relapsing-remitting pattern, and surgery and tumorigenesis are two of its most serious long-term outcomes [1]. Serious disease activity presented with increased levels of inflammatory biomarkers, and aggravated clinical manifestations were commonly attributed to the acute inflammation status. Notably, recurrent inflammatory attacks that cannot be alleviated for a long time, leading to a prolonged active disease phase, are associated with an increased risk of a poor prognosis, including abdominal surgery and tumorigenesis [2]. Dyslipidaemia is a common pathophysiological characteristic that leads to metabolic disorders, altered steroid hormone secretion, and chronic subclinical inflammation [3,4,5]. Indeed, due to alterations in the tumour cell microenvironment and systemic immune status, abnormal lipid metabolism not only affects the inflammatory condition but also affects tumour growth and invasion [3,5,6].

It has been reported that serum lipid profiles differ not only in Eastern and Western populations [7,8,9,10] but also in patients with UC and the general population [11,12]. Remarkably, previous studies have demonstrated that serum lipid abnormalities not only aggravate disease activity in UC [2,13] but are also associated with an increased likelihood of surgery [14] and tumours [15] among patients with IBD, revealing a potential role of dyslipidaemia in inflammation and disease progression. However, only a few studies have assessed the association between lipid profiles and morbidity [16] and disease activity [13] of UC, and they had small sample sizes and inconsistent results. Due to low mortality, long-term outcomes, including tumorigenesis, have rarely been evaluated.

In this study, we comprehensively estimated the correlation between lipid patterns and disease activity as well as long-term prognosis, as represented by surgery-free survival (SFS) and tumour-free survival (TFS), to better distinguish high-risk subgroups and optimize management along with the individualized treatment of UC patients.

## 2. Materials and Methods

### 2.1. Patients

Consecutive patients diagnosed with UC at Peking Union Medical College Hospital (PUMCH) from January 2003 to September 2020 were enrolled in this study. The study enrolment flowchart is shown in Figure 1. The inclusion criteria were as follows: (1) patients diagnosed with UC according to the European Crohn’s and Colitis Organization consensus guidelines [17] combined with data about the clinical presentation, endoscopic features, and imaging or pathological characteristics and (2) those followed-up for at least 6 months. The exclusion criteria were as follows: (1) patients diagnosed with tumours before UC diagnosis; (2) those with missing details about lipid levels, use of medications, and surgery; (3) those with previous autoimmune diseases and those receiving treatment with biologics, immunomodulators, or corticosteroids at baseline; and (4) those who changed the primary diagnosis from UC to other disease or were lost to follow-up. All participants with UC were followed-up (last visit record accessed via the hospital information system (HIS)/telephone follow-up).

### 2.2. Outcome Measures

The long-term outcomes were UC-related surgery and tumorigenesis. Surgery-free survival (SFS) was defined as the period from UC diagnosis to UC-related surgery, and tumour-free survival (TFS) was defined as the period from UC diagnosis to the first tumour occurrence. The end-point events were initial tumour diagnosis, death, date of the last visit, and follow-up cut-off date (30 September 2021), whichever occurred first.

### 2.3. Study Variables

The following data were collected: demographic characteristics, age at diagnosis, the extent of lesions at diagnosis and maximal lesions, medications, the performance of UC-related surgery, complications, extraintestinal manifestations, intestinal infections, colonoscopy record, and lipid profiles. A history of concomitant use of medications was included in the analysis. The development of tumours (including intestinal and other locations) and deaths were reported during follow-up.

### 2.4. Definitions

According to Asian standards, overweight was defined as a body mass index (BMI) of ≥23 kg/m^2^. In line with previous national studies in China, dyslipidaemia was defined as either a total cholesterol (TC) level of ≥6.2 mmol/L, triglyceride (TG) level of ≥2.3 mmol/L, high-density lipoprotein cholesterol (HDL-C) level of <1.0 mmol/L, low-density lipoprotein cholesterol (LDL-C) level of ≥4.1 mmol/L, or self-reported use of lipid-lowering medication, according to the Guidelines for the Prevention and Treatment of Dyslipidaemia in Chinese Adults (2016).

Clinical symptoms and endoscopic findings were selected within one month before or after the lipid measurement. The point lipid measurement was used to evaluate the short-term lipid profiles. The average of multiple lipid measurements (which was used for assessment in grouping based on the severity of dyslipidaemia) was used to evaluate the long-term lipid profiles.

The extent of disease was characterized according to the Montreal classification [18]. According to the 2019 American College of Gastroenterology (ACG) Clinical Guidelines [19], the endoscopic disease activity index (DAI) scoring system includes the Mayo Endoscopic Score (MES) and the Ulcerative Colitis Endoscopic Index of Severity (UCEIS) [20]. The full Mayo Score (MS) containing the endoscopic subscore was used to characterize the clinical DAI.

Biological remission was defined as normal hypersensitive C-reactive protein (hsCRP) level (<3 mg/L) [21], and endoscopic remission was defined as MES ≤ 1 [22]. UC-related surgery was defined as intestinal surgery (mainly including partial or total bowel resection and enterostomy) performed after diagnosis with UC. The severe complications during the disease course were UC-related bowel obstruction/stenosis, bowel perforation, toxic megacolon, abdominal infection, massive gastrointestinal bleeding, and infectious shock.

### 2.5. Statistical Analysis

The Statistical Package for the Social Sciences software version 26 (SPSS Inc., Chicago, IL, USA) and GraphPad Prism software version 9.0 (GraphPad Software, San Diego, CA, USA) were used for data analysis and statistical display. Continuous variables are presented as the mean ± standard deviation or median (interquartile range, IQR), and categorical variables are presented as numbers and proportions. Univariate analyses of qualitative/quantitative differences between groups were conducted using the chi-square test or Fisher’s exact test. Moreover, Student’s *t*-test or the Mann–Whitney U test was used to evaluate continuous variables. Of the patients included, 328 (66.0%) with multipoint lipid measurements (that are ≥3 times higher before the end-point events) were selected for substudy analysis. Patients were classified into the normal group (without abnormal lipid test results), transient dyslipidaemia group (with 1–2 abnormal lipid test results), and persistent dyslipidaemia group (with ≥3 abnormal lipid test results) according to the frequency of abnormal lipid test results before surgery and tumorigenesis. Significant factors in the univariate analyses were included in the multivariate Cox regression analysis and logistic regression analysis. Then, hazard ratios (HRs), odds ratios (ORs), and 95% confidence intervals (CIs) were calculated to determine factors associated with long-term outcomes. Survival analysis was performed using Kaplan–Meier curves. A two-sided *p*-value of < 0.05 was considered statistically significant.

## 3. Results

### 3.1. Clinical Characteristics of Patients

In this study, 497 UC patients who had detailed lipid profiles were eligible, with a male-to-female ratio of 1.23:1 and four (0.8%) deaths. The average age at UC diagnosis was 35.4 (IQR 27.3–46.4) years, and the follow-up duration was 7.5 (IQR 3.8–13.3) years. There were 278 (55.9%) patients diagnosed with dyslipidaemia and 72 (14.5%) patients who experienced severe complications. The patients’ demographic features are shown in Table 1.

We also analysed the impact of lipid levels on baseline clinical characteristics (Table 1). Male sex (62.2% vs. 46.1%, *p* < 0.001), BMI ≥ 23 kg/m^2^ (57.9% vs. 30.6%, *p* < 0.001), a history of smoking (31.3% vs. 19.2%, *p* = 0.002), a maximum E3 classification (88.9% vs. 82.2%, *p* = 0.03), and exposure to aspirin (4.7% vs. 2.7%, *p* < 0.001) were associated with a higher proportion of dyslipidaemia. At disease onset, patients with dyslipidaemia were more likely to report a history of hypertension (16.2% vs. 6.8%, *p* = 0.002), hyperuricaemia (26.3% vs. 18.3%, *p* = 0.035), diabetes (15.8% vs. 5.0%, *p* < 0.001), non-alcoholic fatty liver (9.0% vs. 4.1%, *p* = 0.032), and appendectomy (6.1% vs. 4.1%, *p* = 0.004).

### 3.2. Correlation between Lipid Levels and Disease Activity

Of 497 patients with UC, excluding those receiving lipid-lowering drugs, 408 (82.1%) had complete and detailed data about colonoscopy records, lipid profiles, and hsCRP levels. When the severity of the disease was analysed, patients with dyslipidaemia had greater chances of experiencing both a severe clinical (Mayo score, MS) and endoscopic DAI (not only in the MES but also in the UCEIS) than those with normal lipid levels (all *p* < 0.001, Table 1). Patients in the moderate/severe disease activity phase had a higher proportion of dyslipidaemia, compared to those in the remission/mild disease activity phase (all *p* < 0.001, Figure 2A). Compared with the patients with normal lipid levels, patients with dyslipidaemia had significantly higher levels of hypersensitivity C-reactive protein (hsCRP) (2.28 (0.70–6.43) mg/L vs. 14.05 (3.75–48.88) mg/L, *p* < 0.001, Table 1).

To further analyse the relationship between the lipid patterns and disease severity, participants were grouped according to the full Mayo score (MS), which includes the endoscopic subscore, and they were stratified concurrently based on serum lipid category. Total cholesterol (TC), triglyceride (TG), high-density lipoprotein cholesterol (HDL-C), low-density lipoprotein cholesterol (LDL-C), apolipoprotein A1 (ApoA1), apolipoprotein B, lipoprotein alpha (Lp α), and free fatty acid (FFA) levels were included in the serum lipid profiles.

Based on MS, low TC (*p* < 0.001), HDL-C (*p* < 0.001), and LDL-C (*p* < 0.001) levels were associated with higher risks of severe instead of remission disease activity in the patients with dyslipidaemia (Table 2 and Figure 2B). In the patients with normal serum lipids, only the HDL-C levels showed a significant decreasing trend from remission to severe activity (*p* = 0.02, Figure 2C).

### 3.3. Correlation between Lipid Levels and Prognosis

In the 328 patients with long-term lipid levels, we found that a total of 270 (82.3%) and 164 (50.0%) patients had achieved biological remission and mucosal healing, respectively. Furthermore, the relationship between long-term treatment response and medications as well as long-term lipids levels of patients was also analysed. As shown in the univariate analysis (Appendix A) and the logistic regression analysis (Appendix A), no significant difference was found in the incidence of biologic remission between the patients with persistent dyslipidaemia and non-persistent dyslipidaemia (odds ratio (OR): 0.69, 95% CI: 0.30–1.62, *p* = 0.72). However, the incidence of endoscopic remission decreased in the patients with persistent dyslipidaemia (OR: 0.34, 95%CI: 0.18–0.64, *p* = 0.001). In addition, the exposure of glucocorticoids decreased both the incidence of biologic remission (OR: 0.33, 95%CI: 0.11–0.98, *p* < 0.05) and endoscopic remission (OR: 0.52, 95%CI: 0.28–0.96, *p* = 0.039). There was no significant difference in the incidence of biologic remission and mucosa healing in other drug treatments, including aspirin, 5-ASA, IM, and biologic therapy.

According to the univariate analysis (Table 3), there were 94 (18.9%) patients with UC-related surgery and 53 (10.7%) with tumorigenesis in the cohort of 497 UC patients. Dyslipidaemia was associated with an increased likelihood of experiencing UC-related surgery (23.0% vs. 13.7%, *p* = 0.008) and tumorigenesis (13.3% vs. 7.3%, *p* = 0.031), suggesting a relationship with a worse prognosis in patients with UC.

Of 328 patients who had long-term lipid levels, 83 patients with UC-related surgery and 40 patients with tumours were enrolled in the subanalysis for further assessment of the association between average lipid levels and long-term outcomes. As shown in Appendix A, univariate and multivariate Cox regression analyses demonstrated that persistent dyslipidaemia was significantly identified as an independent risk factor for both surgery (HR (hazard ratio): 3.27, 95% CI (confidence interval): 1.86–5.75, *p* < 0.001) and tumorigenesis (HR: 7.92, 95% CI: 3.97–15.78, *p* < 0.001) among the patients with UC. In the Kaplan–Meier survival analysis, patients with persistent dyslipidaemia had statistically shorter SFS (HR: 3.15, *p* < 0.001, Figure 3A) and TFS (HR: 7.46, *p* < 0.001, Figure 3B) than the transient dyslipidaemia group and normal lipid group.

Patients were also stratified according to lipid compositions to analyse the relationship between lipid patterns and long-term outcomes. As shown in Appendix A and Figure 3C, patients with UC-related surgery had lower TC (4.2 (3.1–5.2) mmol/L vs. 4.6 (4.0–5.2) mmol/L, *p* = 0.01); HDL-C (0.9 (0.7–1.2) mmol/L vs. 1.2 (1.1–1.5) mmol/L, *p* < 0.001); LDL-C (2.3 (1.8–3.1) mmol/L vs. 2.7 (2.2–3.2) mmol/L, *p* = 0.04); and ApoA1 (1.1 (1.0–1.2) g/L vs. 1.4 (1.2–1.6) g/L, *p* < 0.001) levels than those who were surgery-free. In contrast, patients experiencing surgery had a higher TG level than the surgery-free group (1.3 (1.0–2.0) mmol/L vs. 1.2 (0.9–1.6) mmol/L, *p* = 0.012, Figure 3C). Moreover, stratified analysis suggested that patients with tumours presented significantly lower HDL-C (0.9 (0.8–1.1) mmol/L vs. 1.2 (1.0–1.4) mmol/L, *p* < 0.001) and ApoA1 (1.1 (1.0–1.2) g/L vs. 1.3 (1.2–1.6) g/L, *p* < 0.05) levels than those without tumours (Figure 3D).In addition, patients who underwent surgery had higher hsCRP levels than those without surgery (18.0 (3.4–58.5) mg/L vs. 3.0 (0.8–11.2) mg/L, *p* < 0.001, Appendix A). A consistent trend was found in the tumour-bearing group when compared to tumour-free patients (7.0 (2.9–52.0) mg/L vs. 3.6 (0.9–19.0) mg/L, *p* < 0.05, Appendix A). Interestingly, when lipid profiles were analysed, levels of HDL-C and ApoA1 were inversely related to hsCRP levels (all *p* < 0.01, Appendix A).

## 4. Discussion

Our long-term cohort study demonstrated an association between dyslipidaemia and severe disease activity as well as early surgery and tumorigenesis among patients with UC in China. Dyslipidaemia may potentially aggravate acute inflammation and promote disease progression by exacerbating chronic inflammation in UC. Remarkably, severe disease activity and poor long-term outcomes were associated with lower HDL-C levels, suggesting a latent pathogenesis of disease development related to lipid profiles.

In our study, the severity of dyslipidaemia was shown to be positively correlated with not only higher hsCRP levels but also severe disease activity in UC, suggesting a relationship between abnormal lipid metabolism and acute inflammation. A previous study reported a similar result, which focused on the mechanism of the downregulation of the lipolytic enzyme activity of inflammatory cytokines, such as tumour necrosis factor, interleukin-6, and interferon-γ [2,14].

More importantly, changes in lipid profiles were correlated with disease activity in the stratified substudy, and the association remained significant in both the patients with dyslipidaemia and those with normal lipid levels. The findings indicated a potential role played by the specific lipid patterns in the acute inflammatory condition in UC. In fact, studies have reported alterations in lipid compositions with a pattern of low TC and high TG levels among patients with IBD [11,23], and this change was more evident during the active disease phase. Inconsistent with the studies above, a decreased level of HDL-C was found to correlate with higher Mayo scores (MS) in our work. The disparities in lipid patterns could be attributed to the differences in dietary habits and incidence of obesity between China and Western countries and revealed the potential disparities in mechanisms associated with inflammation. It has been reported that acute inflammatory phase disorders of lipoprotein metabolism cause changes in lipid and lipoprotein levels among patients with IBD [24]. The study identified an increased serum TC level in mucosal healing status compared to acute activity in UC, reflecting an improvement in general condition, including nutritional status, because of the resolution of acute inflammation [25]. Moreover, levels of TC and LDL-C were lower in the active IBD than in the healthy participants and were correlated with the systemic inflammatory status [26]. Indeed, low levels of HDL-C are associated with inflammatory and immune diseases because of the anti-inflammatory effects of HDL-C as a part of the innate immune system [27]. These studies have validated the close relationship between abnormal lipid compositions and the acute inflammatory state of IBD. Our work implicated a relationship between decreased levels of HDL-C and acute inflammation in UC, and physiological mechanisms should be explored. Notably, the results suggested that HDL-C could be a predictor of a serious acute inflammatory condition in UC when patients were trapped in a phase of severe disease activity. Since HDL-C levels change more slowly than CRP, HDL-C has a more lasting effect as a surrogate marker of disease severity. Moreover, due to the importance of endoscopy for disease activity assessment, HDL-C level testing might be a substitute in UC patients who do not undergo colonoscopy in clinical work.

Another significant finding of the present study was that persistent dyslipidaemia was independently associated with higher risks of surgery and systemic tumorigenesis in UC, implying a worse prognosis. Furthermore, the lipid patterns were comparable between different groups divided according to prognostic events and levels of hsCRP, suggesting a correlation between specific lipid patterns, long-term outcomes, and chronic inflammation.

There is no denying that abdominal operation is regarded as an important sign of disease progression. Although surgeries in IBD patients are considered to have an impact on lipid metabolism [28], Romanato et al. demonstrated that lipid profiles after intestinal resection were stable, and CD recurrence but not the extent of intestinal resection was the main predictor of changes in lipid profiles [29]. Another study implied that effective prevention of relapse for CD patients might be achieved through moderate dietary fat intake, particularly when the disease condition is unstable [30]. In general, existing studies support that changes in blood lipid levels influence disease development. Studies have reported a lipid pattern of lower TC and higher TG levels in IBD patients with intestinal resections [14,28]. In this work, persistent dyslipidaemia was confirmed as a risk factor for early abdominal surgery in UC, and the level of HDL-C but not the level of TC or TG, as previously described, could be a predictor of bowel resection following diagnosis. Disparities in lipid patterns may have resulted from the location of the operative intestinal segment, inflammatory state, nutritional status, and dietary structure. However, some studies also show that dyslipidaemia aggravates chronic inflammation of the intestine [2], suggesting a potential pathophysiological mechanism between alterations in lipid profiles and disease progression.

Consistent with a previous study [15], our work also showed that dyslipidaemia played a role in the incidence of tumorigenesis associated with colitis. Regarding the mechanism, dyslipidaemia has been validated to prolong the disease course and increase cumulative genetic mutations in the intestinal mucosa, leading to an increased risk of tumorigenesis [2]. Furthermore, dyslipidaemia alters the tumour microenvironment and immunosuppressive function via systemic inflammation, which accelerates tumour progression [15]. However, the reprogramming of lipid metabolism by tumour cells affects serum lipid levels in the body, which are inversely associated with C-reactive protein (CRP) levels in IBD [26,31]. Another study indicated that high TG and LDL-C levels are risk factors for tumorigenesis [32], and applying abnormal lipid patterns is associated with the long-term outcomes of IBD. Similar to our results, studies have reported that serum HDL-C and ApoA1 levels were inversely associated with CRP in IBD patients [26], suggesting a relationship with systemic inflammation. Clearly, low HDL-C levels were not only associated with the incidence of IBD, being [33] inversely related to acute inflammatory conditions [23], but also correlated with chronic inflammation, which may promote tumorigenesis [34,35]. Remarkably, HDL-C participates in both innate and adaptive immune responses [36] and is found to have pleiotropic properties in cell growth, which revealed underlying mechanisms of occurrence and development of tumours [37]. In our study, patients with shorter SFS and TFS were more likely to have lower HDL-C and ApoA1 levels, reflecting a specific lipid pattern related to a worse prognosis. Considering the significantly lower levels in the severe activity phase in our study, low HDL-C levels may be a predictive marker of disease severity and poor prognosis. Consequently, in addition to controlling LDL-C levels emphasized in the existing guidelines, our study demonstrated that the management of HDL-C levels cannot be disregarded in preventing severe disease activity and unfavourable prognosis in UC.

To our knowledge, our cohort is one of the largest cohorts of UC, clarifying the correlation between dyslipidaemia and disease activity in China. Remarkably, this is the first study validating the disparities in lipid profiles of long-term prognosis in UC. A variety of disease activity scoring methods, including clinical DAI, two endoscopic scores, and laboratory indicators (selection of hsCRP), were used to evaluate the inflammatory status of the disease, showing reliability. Importantly, instead of point measurements, the average lipid levels from multiple tests provided a more realistic image of long-term lipid profiles to assess the long-term outcomes. However, this was a retrospective, single-centre observational study. Thus, the continuity of clinical data was compromised and might still be biased despite correction using statistical methods. Finally, we have no data on enteral and parenteral nutrition, diet, or other pharmacological factors, which usually have potential effects on serum lipid levels.

## 5. Conclusions

Our findings revealed a potential pathophysiological mechanism between lipid metabolism and the development of UC. Good management of dyslipidaemia, particularly HDL-C levels, is needed for the early identification of high-risk groups for surgery and prevention of poor outcomes. Therefore, future prospective cohort and laboratory studies should be performed to validate our findings and explore the molecular mechanisms of lipid metabolism and UC progression.

## Figures and Tables

**Figure 1 nutrients-14-03040-f001:**
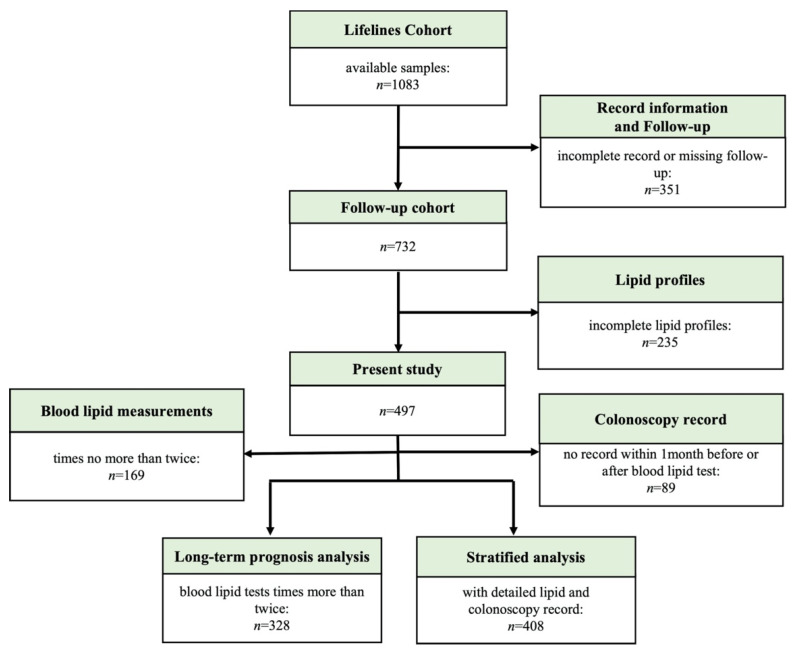
Flowchart of cohort participant inclusion.

**Figure 2 nutrients-14-03040-f002:**
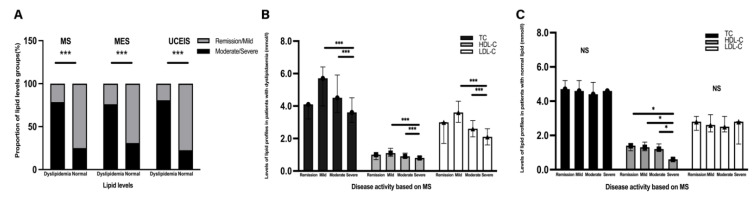
Correlation between lipid levels and disease activity. (**A**) The proportion of patients with dyslipidaemia and normal lipid levels is presented by bar plots and stratified according to disease activity based on MS, MES, and UCEIS. (**B**,**C**) Correlation between lipid profiles and disease activity based on MS in patients with dyslipidaemia and normal lipid levels. (**B**) Bar plots of levels of TC, HDL-C, and LDL-C in different groups based on disease activity in the patients with dyslipidaemia. (**C**) Bar graphs of the levels of TC, HDL-C, and LDL-C in different groups based on disease activity in patients with normal lipid levels. The proportion of lipid levels in patients with dyslipidaemia and normal serum lipid levels is presented by bar plots and stratified according to disease activity. The chi-square test was performed to analyse statistical significance. MS, Mayo score; MES, Mayo Endoscopic Score; UCEIS, Ulcerative Colitis Endoscopic Index of Severity; NS, no significance. Significant differences were adjusted for multiple tests by Bonferroni correction. Significant *p*-value < 0.05. * *p* < 0.05, *** *p* < 0.001.

**Figure 3 nutrients-14-03040-f003:**
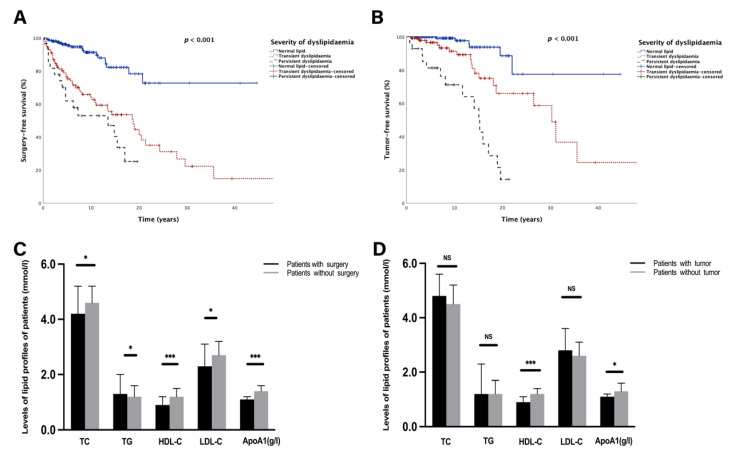
Comparison of lipid levels and lipid profiles in patients with different long-term outcomes. (**A**,**B**) Kaplan–Meier survival curve presenting the surgery-free survival (**A**) and tumour-free survival (**B**) in different groups based on the severity of dyslipidaemia. (**C**) Bar graphs of long-term levels of TC, TG, HDL-C, LDL-C, and ApoA1 between patients who were surgery-free and those who underwent surgery. (**D**) Bar graphs of long-term levels of TC, TG, HDL-C, LDL-C, and ApoA1 between tumour-free and tumour-bearing patients. NS, no significance. Significant *p*-value < 0.05. ** p* < 0.05, **** p* < 0.001. Surgery-free survival was defined as the period from diagnosis to UC-related surgery. Tumour-free survival was defined as the period from diagnosis to tumorigenesis. Transient dyslipidaemia was defined as occurrence of dyslipidaemia not more than twice during the whole following period. Persistent dyslipidaemia was defined as times of dyslipidaemia occurring not less than three times during the whole following period.

**Table 1 nutrients-14-03040-t001:** Demographic and clinical characteristics of patients with dyslipidaemia and normal lipid levels.

Variables *n* (%)	All (*n* = 497)	Dyslipidaemia (*n* = 278)	Normal Lipid (*n* = 219)	*p*-Value *^#^*
Time to follow-up, median years (IQR)	7.5 (3.8–13.3)	7.4 (3.7–13.7)	7.6 (3.8–13.1)	0.27
Age, median years (IQR)	35.4 (27.3–46.4)	37.2 (28.1–47.6)	34.1 (27.2–44.8)	0.91
Male	274 (55.1)	173 (62.2)	101 (46.1)	<0.001 *****
*^a^* Disease duration ≥ 10 years	246 (49.5)	141 (50.7)	105 (47.9)	0.54
BMI ≥ 23 (kg/m^2^)	228 (48.9)	161 (57.9)	67 (30.6)	<0.001 *****
Smoke	129 (25.9)	87 (31.3)	42 (19.2)	
Never	368 (74.0)	191 (68.7)	177 (80.8)	<0.01 ****
Former	100 (20.1)	65 (23.4)	35 (16.0)	
Current	29 (5.9)	22 (7.9)	7 (3.2)	
Maximum extent diagnosed				
E1	10 (2.0)	4 (1.4)	6 (2.7)	0.482 ^b^
E2	60 (12.1)	27 (9.7)	33 (15.1)	0.069
E3	427 (85.9)	247 (88.9)	180 (82.2)	0.03 ***
Comorbidities				
Hypertension	60 (12.1)	45 (16.2)	15 (6.8)	<0.01 ****
Hyperuricemia	113 (22.7)	73 (26.3)	40 (18.3)	0.04 ***
Diabetes	55 (11.1)	44 (15.8)	11 (5.0)	<0.001 *****
NAFLD	34 (6.8)	25 (9.0)	9 (4.1)	0.03 ***
Appendectomy	40 (8.0)	31 (6.1)	9 (4.1)	<0.01 ****
Severe complications	72 (14.5)	43 (15.5)	29 (13.2)	0.48
EIMs	75 (15.1)	47 (16.9)	28 (12.8)	0.20
Medications				
Glucocorticoid	402 (80.9)	232 (83.5)	170 (77.6)	0.10
IM	165 (33.2)	98 (35.3)	67 (30.6)	0.27
Biological therapy	75 (15.1)	39 (14.0)	36 (16.4)	0.46
5-ASA	474 (95.4)	267 (96.0)	207 (94.5)	0.76
Aspirin	19 (3.8)	13 (4.7)	6 (2.7)	<0.001 *****
DAI (*n* = 408)				
MS				
Remission	43 (8.7)	2 (0.9)	41 (22.9)	<0.001 *****
Mild	101 (20.3)	29 (12.7)	72 (40.2)	
Moderate	170 (34.2)	107 (46.7)	63 (35.2)	
Severe	94 (18.9)	91 (39.7)	3 (1.7)	
MES				
Remission	51 (10.3)	9 (3.9)	42 (23.5)	<0.001 *****
Mild	66 (13.3)	19 (8.3)	47 (26.3)	
Moderate	108 (21.7)	63 (27.5)	45 (25.1)	
Severe	183 (36.8)	138 (60.3)	45 (25.1)	
UCEIS				
Remission	45 (9.1)	3 (1.3)	42 (23.5)	<0.001 *****
Mild	105 (21.1)	26 (11.3)	79 (44.1)	
Moderate	168 (33.8)	111 (48.5)	57 (31.8)	
Severe	90 (18.1)	89 (38.9)	1 (0.6)	
hsCRP, median mg/L (IQR) (*n* = 408)	5.28 (1.42–28.79)	14.05 (3.75–48.44)	2.28 (0.70–6.43)	<0.001 *****

Statistics are performed using linear regression for continuous variables and chi-square test for categorical variables. When appropriate, values are reported as median (IQR) or number (%). *^#^* Comparison between dyslipidaemia and normal lipid groups. *^a^* Disease duration: time from onset to follow up ≥ 10 years. ^b^ Continuity correction. IQR, interquartile range; BMI, body mass index; NAFLD, non-alcoholic fatty liver; EIMs, extraintestinal manifestations; 5-ASA, 5-aminosalicylic acid; IM, immunosuppressant; hsCRP, hypersensitivity C-reaction protein; MS, Mayo score; MES, Mayo Endoscopic Score; UCEIS, Ulcerative Colitis Endoscopic Index of Severity. Significant *p*-value < 0.05. ** p* < 0.05, *** p* < 0.01, **** p* < 0.001.

**Table 2 nutrients-14-03040-t002:** Comparison of lipid profiles between different groups of disease activity based on MS in patients with dyslipidaemia and normal lipid levels.

Laboratory Tests	Dyslipidaemia (*n* = 229)				*p*-Value	Normal Lipid (*n* = 179)				*p*-Value
Median (IQR)	Remission (*n* = 2)	Mild (*n* = 29)	Moderate (*n* = 107)	Severe (*n* = 91)		Remission (*n* = 41)	Mild (*n* = 72)	Moderate (*n* = 63)	Severe (*n* = 3)	
TC (mmol/L)	4.1 (3.2-)	5.7 (4.0–6.4)	4.5 (3.6–5.9)	3.6 (3.0–4.5)	<0.001 *****	4.7 (4.1–5.2)	4.6 (4.0–5.2)	4.4 (4.0–5.1)	4.6 (3.4-)	0.69
TG (mmol/L)	1.5 (1.2-)	1.4 (1.1–1.8)	1.4 (1.1–2.4)	1.2 (0.8–1.8)	0.06	1.2 (0.9–1.6)	1.2 (1.0–1.4)	1.2 (1.0–1.4)	1.1 (0.8-)	0.85
HDL-C (mmol/L)	1.0 (0.7-)	1.1 (0.9–1.4)	0.9 (0.8–1.1)	0.8 (0.7–0.9)	<0.001 *****	1.4 (1.1–1.5)	1.3 (1.1–1.6)	1.2 (1.1–1.5)	0.6 (0.5-)	0.02 ***
LDL-C (mmol/L)	3.0 (1.7-)	3.6 (3.0–4.3)	2.6 (2.1–3.1)	2.1 (1.6–2.6)	<0.001 *****	2.8 (2.3–3.1)	2.6 (2.2–3.2)	2.5 (2.2–3.1)	2.8 (1.5-)	0.77
ApoA1 (g/L)	1.3 (1.2-)	1.2 (1.1–1.4)	1.2 (1.0–1.5)	1.0 (0.8–1.2)	0.09	1.2 (1.2–1.8)	1.5 (1.2–1.8)	1.3 (1.2–1.6)	1.2 (1.1-)	0.30
ApoB (mg/L)	1.0 (0.8-)	1.1 (1.0–1.4)	1.0 (0.8–1.2)	0.9 (0.7–1.0)	0.07	0.9 (0.6–0.9)	0.8 (0.7–1.0)	0.8 (0.7–1.0)	1.0 (0.7-)	0.94
Lp α (mg/L)	96.0 (82.0-)	82.0 (70.0–277.0)	205.0 (71.5–428.0)	102.5 (32.0–188.3)	0.12	136.0 (25.0–211.8)	120.0 (53.0–248.0)	101.0 (70.8–144.5)	171.0 (101.0-)	0.96
FFA (umol/L)	453.5 (402.0-)	551.0 (463.0–560.0)	551.0 (338.8–656.5)	481.5 (322.5–618.3)	0.69	551.0 (222.0–698.8)	400.0 (312.5–802.5)	470.5 (322.8–722.5)	634.0 (567.0-)	0.99
hsCRP (mg/L)	2.1(1.9-)	4.6 (1.2–36.5)	9.4 (1.8–44.0)	20.5 (11.6–55.4)	<0.01 ****	2.8 (1.4–4.5)	1.6 (0.5–6.8)	2.2 (0.8–5.7)	35.2 (1.3-)	0.22

IQR, interquartile range; TC, total cholesterol; TG, total triglyceride; HDL-C, high-density lipoprotein cholesterol; LDL-C, low-density lipoprotein cholesterol; ApoA1, apolipoprotein A1; ApoB, apolipoprotein B; Lp α, lipoprotein α; FFA, free fatty acid; hsCRP, hypersensitivity C-reaction protein; MS, Mayo score. Significant *p*-value < 0.05. ** p* < 0.05, *** p* < 0.01, **** p* < 0.001.

**Table 3 nutrients-14-03040-t003:** Correlation between dyslipidaemia and poor prognosis in UC patients.

Variables *n* (%)	All (*n* = 497)	Dyslipidaemia (*n* = 278)	Normal Lipid (*n* = 219)	*p*-Value *^#^*
UC-related surgery	94 (18.9)	64 (23.0)	30 (13.7)	< 0.01 **
Tumorigenesis	53 (10.7)	37 (13.3)	16 (7.3)	0.03 *

*^#^* Comparison between dyslipidaemia and normal lipid groups. Significant *p*-value < 0.05. ** p* < 0.05, *** p* < 0.01.

## Data Availability

The data underlying this study are available from the corresponding author upon reasonable request.

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
