# Peer review of "Dyslipidaemia Is Associated with Severe Disease Activity and Poor Prognosis in Ulcerative Colitis: A Retrospective Cohort Study in China"

_nutrients, 2022, doi:10.3390/nu14153040_

Round 1
Reviewer 1 Report
The manuscript entitled “Dyslipidaemia is associated with severe disease activity and poor prognosis in ulcerative colitis: A retrospective cohort study in China” by Liu et al. showed the persistant dyslipidaemia associated t with serious risk disease activity and worse long term outcomes in ulcerative colitis (UC) patients. They assessed a cohort of 497 patients and performed long term follow up study. This study is carefully designed and highlight the dyslipidimia could be the predictive feature of future complications in the UC patients.
Comment:
1. Dyslipidaemia and disease location/classification: Does dyslipidaemia corelates with disease location or types of ulcerative colitis.
Minor comment:
1. It is extremely difficult to read axis labels in Figure 2
Author Response
Point 1: Dyslipidaemia and disease location/classification: Does dyslipidaemia corelates with disease location or types of ulcerative colitis.
Response 1: Thank you for your detailed comments. In Table 1, we have analyzed the relationship between disease location and dyslipidaemia. However, due to the need to refine the table length, only the correlation between E3 (i.e. extensive colonic type) disease locationand lipid levels was displayed, which suggested a significant difference (p < 0.01). The results showed that the proportion of maximum extent diagnosed with E3 was higher in patients with dyslipidaemia than those with normal lipid (p=0.08). However, no significant difference was found between dyslipidaemia and E1 as well as E2 classification. And there was no significant difference between dyslipidaemia and disease location in the whole cohort (p=0.103). The detailed analysis of the relationship between lesion location and lipid levels has been added to the revised table 1 in the revised manuscript. Here, the statistical chart of correlation between dyslipidaemia and disease location is attached. Thank you again for your constructive comments.
Part of Revised Table 1. correlation between dyslipidaemia and disease location in patients with UC
|
Variables n (%) |
All (n=497) |
Dyslipidaemia (n=278) |
Normal lipid (n=219) |
P value# |
|
|
|
Maximum extent diagnosed |
||||||
|
E1 |
10(2.0) |
4(1.4) |
6(2.7) |
0.482b |
0.103 |
|
|
E2 |
60(12.1) |
27(9.7) |
33(15.1) |
0.069 |
||
|
E3 |
427(85.9) |
247(88.9) |
180(82.2) |
0.03* |
||
#Comparison between dyslipidaemia and normal lipid groups. b Continuity correction.
Point 2: It is extremely difficult to read axis labels in Figure 2.
Response 2: Thank you for your positive comments. We feel extremely sorry for the difficulty in your review due to our unclear axis labels. In Figure 2, we have analyzed the correlation between lipid levels and disease activity. First, the proportion of patients with dyslipidaemia and normal lipid levels is presented by bar plots and stratified according to disease activity based on MS, MES, and UCEIS in Figure 2A. Next, in Figure 2B and Figure 2C, the correlation between lipid profiles and disease activity based on MS in patients with dyslipidaemia and normal lipid levels were shown respectively. Due to the limitation of image pixels and the problem of image font size during the upload manuscript, the axis labels are too small to read clearly. According to your suggestion, we have simplified the axis labels and resized the font size to make it easier to read. The revised Figure 2 has been adjusted in the revised manuscript. The revised Figure 2 is also attached here (See the attached word document). Thank you again for pointing out the insufficiency of our figure to make our manuscript more perfect.
Figure legend:
Figure 2. Correlation between lipid levels and disease activity. (A) The proportion of patients with dyslipidaemia and normal lipid levels is presented by bar plots and stratified according to disease activity based on MS, MES, and UCEIS. (B-C) Correlation between lipid profiles and disease activity based on MS in patients with dyslipidaemia and normal lipid levels. (B) Boxplots of levels of TC, HDL-C, and LDL-C in different groups based on disease activity in the patients with dyslipidaemia. (C) Bar graphs of the levels of TC, HDL-C, and LDL-C in different groups based on disease activity in patients with normal lipid levels. The proportion of lipid levels in patients with dyslipidaemia and normal serum lipidlevels is presented by bar plots and stratified according to disease activity. The chi-square test was performed to analyze statistical significance.MS= Mayo score. MES = Mayo endoscopic score. UCEIS= Ulcerative colitis endoscopic index of severity. NS= No significance. Significant differences were adjusted for multiple tests by Bonferroni correction. Significant p value <0.05. *p<0.05,***p<0.001.

Reviewer 2 Report
This is a long-term cohort study investigating the association of dyslipidaemia
with disease activity and prognosis in patients with ulcerative colitis. Liu Z et . al pointed out the dyslipidaemia, and particularly altered HDL-C levels, as a high risk for surgery in UC patients and UC progression. Although the story is not novel, this study corroborates the potential link between lipid metabolism and the development of UC.
The study is clearly written and the statistical methods appropriate for such a study appropriate. However, the authors should include the impact of response to therapy in these patients. Drug treatments received over time should be also recorded and considered in a subsequent analysis. Since a subgroup of UC (56,1% ) patient present Dyslipidemia, which is a potential reason explaining this association with the disease? How is the % of non-responders in this subgroup? The patients reached the remission are just 2, none conclusion can be done on these patients.
Author Response
Point 1: The study is clearly written and the statistical methods appropriate for such a study appropriate. However, the authors should include the impact of response to therapy in these patients. Drug treatments received over time should be also recorded and considered in a subsequent analysis. Since a subgroup of UC (56,1% ) patient present Dyslipidemia, which is a potential reason explaining this association with the disease? How is the % of non-responders in this subgroup? The patients reached the remission are just 2, none conclusion can be done on these patients.
Response 1: Thank you very much for your detailed comments and insightful suggestion. In the Materials and methods section, we have detailed clarified that the point lipid measurement was used to evaluate the short-term outcome(disease activity) and the average of multiple lipid measurements (which was used for assessment in grouping based on the severity of dyslipidaemia) was used to evaluate the long-term outcomes(surgery-free survival [SFS] and tumour-free survival[TFS]). In the result part of 3.2Correlation between lipid levels and disease activity, the patients reached the remission were just 2, which actually showed a correlation with point lipid level and disease activity. However, as you pointed out, no conclusion about correlation between blood lipids and long-term prognosis can be done on these patients. Therefore, in this study, we also used the long-term blood lipid levels for further analysis on long-term prognosis, making the study more reliable.
We agree with your suggestion very much that drug treatments received over time should be recorded and considered in a subsequent analysis. In fact, we have collected this part of information about patients. As shown in Supplementary Table S3(former Supplementary Table S1), univariate and multivariate Cox regression analyses demonstrated that medications were not significantly identified as independent risk factors for both surgery and tumorigenesis among the patients with UC. According to your suggestion, we have supplemented the analysis of the relationship between treatment response and medications as well as long-term lipids levels of patients. In the revised supplemental materials, biological remission and endoscopic remission as outcomes, the correlation analysis of drug treatments, long-term average blood lipid levels and treatment response was analyzed. Biological remission was defined as normal hypersensitive C-reactive protein (hsCRP) level (< 3mg/l)(1), and endoscopic remission was defined as Mayo endoscopic score ≤ 1(2), as shown in the revised Materials and methods section.
Results as shown in the revised supplementary table S1 , in the 328 patients with long-term blood lipid levels, we found that a total of 270(82.3%) and 164(50.0%) patients had achieved biological remission and mucosal healing, respectively. In the univariate analysis(revised supplementary table S1), compared with patients without biological remission, the patients who had achieved biological remission had lower proportion of BMI≥23 (35.2% vs. 55.2%, p = 0.005), use of 5-ASA(92.6% vs. 100.0%, p = 0.03) and glucocorticoids (77.8% vs. 93.1%, p = 0.01). In addition, patients who had endoscopic remission had lower proportion of persistent dyslipidaemia (11.0% vs. 25.0%, p = 0.001) and exposure to glucocorticoids (75.6% vs. 85.4%, p = 0.03), compared to patients without endoscopic remission.
In the logistic regression analysis(revised supplementary table S2), the results showed that, no significant difference was found in the incidence of biologic remission between the patients with persistent dyslipidaemia and non-persistent dyslipidaemia (odds ratio [OR]: 0.69, 95% confidence interval [CI]: 0.30-1.62, p= 0.72). However, the incidence of endoscopic remission decreased in the patients with persistent dyslipidaemia (OR: 0.34, 95%CI: 0.18-0.64, p = 0.001). In addition, the exposure of glucocorticoids decreased both the incidence of biologic remission(OR: 0.33, 95%CI: 0.11-0.98, p < 0.05) and endoscopic remission (OR: 0.52, 95%CI: 0.28-0.96, p = 0.039). There was no significant difference in the incidence of biologic remission and mucosa healing in other drug treatments including aspirin, 5-ASA, IM, and biologic therapy.
We have supplemented this part of analysis content in the revised manuscript and added the result table to the supplementary materials(Supplementary Table S1 and Supplementary Table S2). The supplemental statistical charts are also attached here. Thank you again for your constructive comments, which makes our results plumper, more reliable and perfect.
Supplementary Table S1. Comparison of clinical features in the patients with biological remission and endoscopic remission
|
Variables(%), (n=328) |
|
Biological remission(n=270) |
P value |
  |
Endoscopic remission(n=164) |
P value |
|
|
||||||
|
Sex(male) |
|
142(52.6) |
0.55 |
82(50.0) |
0.51 |
|
|
a Disease duration |
|
137(50.7) |
0.73 |
86(52.4) |
0.44 |
|
|
BMI≥23(kg/m2) |
|
95(35.2) |
0.00** |
66(40.2) |
0.57 |
|
|
EIMs |
|
32(76.2) |
0.27 |
22(13.4) |
0.74 |
|
|
Persistent dyslipidaemia |
|
51(18.9) |
0.36 |
18(11.0) |
0.00** |
|
|
Medications |
|
|||||
|
Aspirin |
|
9(3.3) |
1.00c |
6(3.7) |
0.75b |
|
|
5-ASA |
|
250(92.6) |
0.03*c |
153(93.3) |
0.64 |
|
|
Glucocorticoids |
|
210(77.8) |
0.01*b |
124(75.6) |
0.03* |
|
|
IM |
|
89(33.0) |
0.22 |
55(33.5) |
0.73 |
|
|
Biological therapy |
|
42(15.6) |
0.52 |
  |
24(14.6) |
0.45 |
Statistics are performed using linear regression for continuous variables and Chi-squared test for categorical variables. When appropriate, values are reported as median [IQR] or number (%). IQR= interquartile range. a Disease duration: time from onset to follow up ≥ 10 years. BMI = Body Mass Index. EIMs=Extraintestinal manifestations. 5-ASA=5-Aminosalicylic Acid. IM= Immunosuppressant. b Continuity correction. c Fisher's test. Significant p-value <0.05. *p<0.05, **p<0.01, ***p<0.001.
Supplementary Table S2. Logistic regression analysis in the patients with biological remission and endoscopic remission
|
Variables |
Biological remission(n=270) |
  |
Endoscopic remission(n=164) |
||||
|
P value |
OR |
95%CI |
  |
P value |
OR |
95%CI |
|
|
Sex(male) |
0.40 |
1.30 |
0.70-2.38 |
0.43 |
0.83 |
0.52-1.32 |
|
|
a Disease duration |
0.47 |
1.25 |
0.68-1.27 |
0.23 |
1.33 |
0.84-2.08 |
|
|
BMI≥23(kg/m2) |
0.006** |
0.43 |
0.23-0.78 |
0.29 |
0.77 |
0.81-2.08 |
|
|
EIMs |
0.80 |
0.69 |
0.31-1.54 |
0.87 |
1.05 |
0.53-2.04 |
|
|
Persistent dyslipidaemia |
0.72 |
0.69 |
0.30-1.62 |
0.001* |
0.34 |
0.18-0.64 |
|
|
Medications |
|||||||
|
Aspirin |
0.71 |
1.52 |
0.17-12.50 |
0.75 |
1.25 |
1.56-5.56 |
|
|
5-ASA |
1.00 |
0.01 |
0.00- |
0.68 |
0.81 |
0.29-2.22 |
|
|
Glucocorticoids |
0.046* |
0.33 |
0.11-0.98 |
0.039* |
0.52 |
0.28-0.96 |
|
|
IM |
0.60 |
0.84 |
0.44-1.59 |
0.68 |
1.11 |
0.67-1.85 |
|
|
Biological therapy |
0.84 |
1.09 |
0.49-2.44 |
  |
0.45 |
0.78 |
0.40-1.49 |
a Disease duration: time from onset to follow up ≥ 10 years. BMI = Body Mass Index. EIMs=Extraintestinal manifestations. 5-ASA=5-Aminosalicylic Acid. IM= Immunosuppressant. OR=Odds Ratio. 95% CI = 95% confidence interval. Significant p-value <0.05. *p<0.05, **p<0.01, ***p<0.001.
References
- Verstockt B, Mertens E, Dreesen E, Outtier A, Noman M, Tops S, et al. Influence of Drug Exposure on Vedolizumab-Induced Endoscopic Remission in Anti-Tumour Necrosis Factor [TNF] Naïve and Anti-TNF Exposed IBD Patients. J Crohns Colitis. 2020;14(3):332-41.
- Feagan BG, Rutgeerts P, Sands BE, Hanauer S, Colombel JF, Sandborn WJ, et al. Vedolizumab as induction and maintenance therapy for ulcerative colitis. N Engl J Med. 2013;369(8):699-710.
